# Evaluation of the Impact of Cold Atmospheric Pressure Plasma on Soybean Seed Germination

**DOI:** 10.3390/plants10010177

**Published:** 2021-01-19

**Authors:** Renáta Švubová, Ľudmila Slováková, Ľudmila Holubová, Dominika Rovňanová, Eliška Gálová, Juliána Tomeková

**Affiliations:** 1Faculty of Natural Sciences, Comenius University in Bratislava, Mlynská dolina, Ilkovičova 6, 842 15 Bratislava, Slovakia; ludmila.slovakova@uniba.sk (Ľ.S.); ludka.holub@gmail.com (Ľ.H.); rovnanova.d@gmail.com (D.R.); eliska.galova@uniba.sk (E.G.); 2Faculty of Mathematics, Physics and Informatics, Comenius University in Bratislava, Mlynská dolina 6280, 842 48 Bratislava, Slovakia; juliana.tomekova@gmail.com

**Keywords:** Cold Atmospheric Pressure Plasma (CAPP), DNA damage, dehydrogenases, germination, soybean

## Abstract

The present study aims to define the effects of Cold Atmospheric Pressure Plasma (CAPP) exposure on seed germination of an agriculturally important crop, soybean. Seed treatment with lower doses of CAPP generated in ambient air and oxygen significantly increased the activity of succinate dehydrogenase (Krebs cycle enzyme), proving the switching of the germinating seed metabolism from anoxygenic to oxygenic. In these treatments, a positive effect on seed germination was documented (the percentage of germination increased by almost 20% compared to the untreated control), while the seed and seedling vigour was also positively affected. On the other hand, higher exposure times of CAPP generated in a nitrogen atmosphere significantly inhibited succinate dehydrogenase activity, but stimulated lactate and alcohol dehydrogenase activities, suggesting anoxygenic metabolism. It was also found that plasma exposure caused a slight increment in the level of primary DNA damage in ambient air- and oxygen-CAPP treatments, and more significant DNA damage was found in nitrogen-CAPP treatments. Although a higher level of DNA damage was also detected in the negative control (untreated seeds), this might be associated with the age of seeds followed by their lower germination capacity (with the germination percentage reaching only about 60%).

## 1. Introduction

Soybean is an important crop that is a part of the diet in many countries. According to the statista.com portal, around 350 million tons was grown worldwide in the years 2019/2020 (https://www.statista.com/statistics/263926/soybean-production-in-selected-countries-since-1980/). Whereas soybean contains large amounts of B vitamins, essential elements, proteins and lipids, soy consumption is a unique source of nutrients for human and animals. This unique nutritional composition makes it a superfood that is grown worldwide. Due to the reduced germination of soybean seeds, large areas of arable land are used for its cultivation. For this reason, it is necessary to look for new, ecologically and economically advantageous solutions that would ensure better germination of seeds of economically important crops and thus a higher yield on a smaller area. An effective and promising method seems to be the application of non-thermal plasma in agricultural practice [1,2,3]. In laboratory conditions, plasma is generated by ionization of molecule, atoms in different working gases by heating or applying of electric field. Non-thermal plasma (or Cold Plasma) for gentle plasma treatment of biological materials can be easily produced at reduced pressure; however, the need for vacuum equipment is a limiting factor for application of low-pressure plasma in continuous in-line plasma treatment and for low-pressure sensitive substrates. Composition of non-thermal plasma, depending on experimental conditions, is formed by different kind of reactive species, electrons, ions, UV-radiation, radicals and metastable particles and gaseous products [4,5]. Non-thermal plasma generated at atmospheric pressure, referred to as Cold Atmospheric Pressure Plasma (CAPP), is suitable for practical applications due to the easier generation of plasma without the use of low-pressure equipment. Using Fourier Transform Infrared (FTIR) Spectroscopy, it was shown that NO_2_, N_2_O, NO, HNO_2_ are present in CAPP of Diffuse Coplanar Surface Barrier Discharge (DCSBD) generated in ambient air. In the nitrogen atmosphere, small amounts of N_2_O were observed because of oxygen impurities. In the oxygen atmosphere, only the presence of ozone was recorded [6]. Treatment of seed surfaces by non-thermal plasma generated in different gases effectively reduces the amount of pathogenic microorganism, which colonize corn grains [7], nuts [8,9], barley and wheat grains [1], and pine seeds [10]. Non-thermal plasma modifies seed surface, which markedly changes its affinity to water, and this is very important for successful imbibition and starting of germination processes. For example, application of CAPP generated in ambient air expressively influenced, due to the surface oxidation, the wettability and water uptake by wheat, lentils and bean seeds [11]. The study of Nishime et al. [12] confirmed a remarkable increase in wettability (82% reduction of water contact angle) and acceleration in germination (60% faster after 24 h) after treatment of winter wheat grains with a dielectric barrier discharge (DBD). In consequence of these factors, the germination can be significantly accelerated, e.g., via faster activation of lytic enzymes [13,14]. On the other hand, through the disruption of the seed envelope, the reactive particles can penetrate and interact directly with storage tissues and embryo which can damage mainly embryo. The degree of damage/hardness of the seeds is affected by moisture in plant seed material and by seed coat thickness [15]. The combination of reactive species and faster water imbibition can cause the damage of membranes and genetic material [16,17], start fermentation processes and lead to embryo death [18]. It can be summarized that pre-sowing treatment of seeds with CAPP can have a beneficial influence on the germination and growth of young seedlings. It was observed that it increases the rate of germination, growth parameters and biomass production [19,20,21,22], stimulates adaptive responses, and reduces DNA damage [23]. The positive or negative impact on germination, growth and development is species specific and strongly depends on type of plasma and time of application [14]. This is presented in the work by Šerá et al. [24], who confirmed (using two-way ANOVA) a significant influence of both tested factors (type of plasma source, exposure time, and both factors together) on seed germination and growth of young seedlings. It could be added that the plasma effect is also species specific.

The main goal of our work was to find the most suitable pre-sowing CAPP treatment in order to increase the germination of soybean seeds. This study reports novel data regarding the effect of CAPP on the activity of dehydrogenases and lytic enzymes that affect the whole process of seed germination (not just in soybean). This study shows that the CAPP treatment causes slight increment of primary DNA damage (detected by the comet assay) with increasing exposure time; however, it does so without significantly affecting the vigour of the seedlings. CAPP generated in nitrogen for longer treatment times (90 and 120 s) caused the highest DNA damage and these seeds did not germinate.

## 2. Results and Discussion

After treating the dry soybean seeds (*Glycine max* L. cv. Nížina) with CAPP, differences in water uptake were noticed. Water intake and thus seed weight increased linearly with increasing plasma dose, this trend was the most pronounced in the case of plasma generated in a nitrogen atmosphere. Similar results are reported by [25], who showed a significant influence of non-thermal plasma treatments on soybean seed water consumption during imbibition. Many authors declare that increased seed water intake is closely related to plasma surface modifications of seed surface caused by higher doses of plasma [14,20,26,27]. In our study, the highest (statistically significant) water intake by seeds was measured in N60, N90, N120 and O90, O120 variants (Figure 1).

Increased water intake by seeds can be beneficial and may lead to faster germination. This is confirmed by a study [11] stating that non-thermal radiofrequency plasma generated in the ambient air significantly influences wettability and imbibition rate of wheat, lentil and bean seeds, resulting in a significant acceleration of germination. Additionally, application of helium plasma discharge in the power of 60, 80, 100 and 120 W for 5 s on soybean seeds had a positive effect on germination and growth parameters [28]. At our chosen plasma doses, inhibition of soybean seed germination in N90 and N120 variants was noticed. Conversely, rapid (inadequate) water absorption can cause a reduction in the rate of germination or suffocation of the embryo. This is stated, for example, in [11], where the use of CAPP generated in a nitrogen atmosphere for 180 and 300 s led to a significant increase in water intake, high lactate and alcohol dehydrogenase activities and thus to a complete inhibition of pea seed germination. The study presented in [6] confirmed that the generation of DCSBD plasma, used in our work, in a nitrogen and ambient air atmosphere produces ultraviolet (UV) radiation which is harmful in high doses, as confirmed by the work [29]. Nitrogen plasma produces a greater deal of intensive UV radiation than ambient air plasma what could be the reason of low germination of seeds treated by nitrogen plasma at 90 s and 120 s. In other treated variants, an increase in percentage of germination (compared to the untreated control) was recorded, statistically significant increase was in the variants N30, O60, O90. Central Control and Testing Institute in Agriculture in Bratislava, Slovakia who provided soybean seeds in 2017 declared the germination of soybean seeds up to 90% when stored correctly. The seeds were stored in the dark, at a temperature of 8 °C, which corresponds to the correct storage conditions. In 2017 it was really around 87%, but in January 2019, untreated soybean seeds (untreated control) had a relatively low percentage of germination (about 60%), despite suitable storage conditions, probably due to seed aging (see Section 3 Materials and Methods, Section 3.1 Plant Material). The positive effect of CAPP on seed vigour (Seed Vigour Index %), seedling vigour (Seedling Vigour Index %) and also on the Seedling length (cm), Seedling length index (%) was also monitored (Figure 2). As in our study, the beneficial effect of plasma on germination and growth and development of young seedlings (length and weight of seedlings, biomass production) was confirmed for many species of agriculturally important crops—wheat, oat, tomato, mungo bean, pea [14,26,30,31,32,33,34] and it seems to have a use in agricultural practice in order to increase the germination of older seeds as well.

The appearance and vitality of the three-day-old soybean seedlings is documented in Figure 3A. After visualization of ROS, we found that most of the superoxide radical was present in the seed coat and cotyledons in O60, O90, N30—120 variants. In other treatments, ˙O_2_¯ was mainly in the root tip. In the case of hydrogen peroxide, brownish spots on cotyledons were recorded and it was mostly found in the root tip and elongation zone of the root (Figure 3B,C). ROS are produced normally in plants during growth and development and are effectively detoxified by antioxidant enzymes. The balance between production and detoxification of ROS may be disturbed by exposure to stronger environmental factors [36,37]. Tomeková et al. [6], using FTIR spectroscopy, pointed to production of ozone (O_3_) during CAPP generation in oxygen atmosphere and nitrogen oxides during CAPP generation in ambient air. Like UV radiation, long-term exposure to O_3_ or reactive nitrogen species leads to oxidative stress and to partial or permanent disruption of homeostasis [38,39]. The accumulation of ˙O_2_^−^ and H_2_O_2_ in the root tips and the elongation zone of the roots of soybean seedlings is associated with intensive cell division and elongation, thus having a physiological basis. Their excessive accumulation in cotyledons (especially in variants N90, N120) and subsequent inhibition of germination indicates pathological ROS production and failure of enzymatic and non-enzymatic detoxification defence mechanisms.

The action of CAPP, as a reactive environment, certainly causes increased oxidative stress. Oxidative stress can lead to irreversible damage of membranes, but this depends on how effectively the antioxidant system works. We confirmed the accumulation of ROS by monitoring the activities of SOD that normally detoxifies ˙O_2_^−^ to H_2_O_2_ [40] and G-POX, which decomposes the generated H_2_O_2_ with the help of a donor of two electrons, into water. Statistically significant increases in activity of SOD were observed in O60, O120, A30 and A60 variants. In other CAPP-treated seedlings, the activity of SOD was comparable with the untreated control. The highest G-POX activity, compared to the untreated control, was in N30, N60, O120 and A120 treatments (Figure 4). In the case of N90 and N120, antioxidant enzymes apparently did not manage to capture the emerging ROS, which led to the killing of the embryo. These assumptions are also confirmed by the high amount of malondialdehyde (MDA) in N90 and N120 variants, which indicates a significant lipid peroxidation (Appendix A). Similar data are presented in the study by [14].

Soy represents the most efficient source of plant proteins. Detailed study of soybean seed storage proteins was carried out, and more than 80% of the proteins identified in soybean cotyledons were subunits of glycinin and β-conglycinin, two major storage proteins [41]. During germination and early growth of the seedling, storage proteins are degraded and mobilized by proteases, but the degradation of β-conglycinin during storage protein mobilization appeared to be similar to that of glycinin but at a faster rate [42]. These solubilized proteins can be quantified by various methods, for example, according to Bradford [43]. Due to CAPP treatment, the concentration of total soluble protein increased significantly in A90 variant, which is probably related to the significant increase of protease activity. In other variants, the number of proteins was comparable to the control. However, we noticed changes in the level of activity of lytic enzymes, which cleave storage substances in soybean cotyledons. The activity of the protease, an enzyme that cleaves proteins, was affected. A slight increase in protease activity in O30, A30 variants and a statistically significant increase in A90 and O120 variants was measured. Because soybean contains practically no starch, amylase activity was very low. Therefore, the significant increase in its activity in the variants A60, O90 and O120 was quite surprising. Glucanase activity was lower in all CAPP-treated variants (except N30) compared to the untreated control, see on Figure 5. The effect of non-thermal plasma treatment on total protein content and lytic enzymes activities in pea (*Pisum sativum* L. cv. Prophet) was studied in [14]. The authors state that non-thermal plasma generated in air and nitrogen atmosphere at treatment time of 60 s had a significant positive effect on lytic enzymes activities. Sadhu et al. [13] studied the influence of non-thermal plasma on the enzymatic activity in germinating mung beans and reported an increase in soluble sugar and protein concentrations, as well as an increase in hydrolytic enyme activity like amylase, protease and phytase after the treatment.

Dehydrogenases are enzymes from the oxidoreductase family that catalyse the removal of hydrogen from the substrate which is oxidized. They affect a large number of processes, including seed germination. Alcohol (ADH) and lactate (LDH) dehydrogenases operate in anoxygenic metabolism, use alcohols and lactate as substrates and in this way the young seedlings gain energy in the early stages [44]. Plasma significantly affected the activity of these enzymes. A significant and statistically significant increase in O30, A60, N60, A90, O120 was recorded. The activities of LDH and ADH in non-germinating, imbibed seeds (embryo + cotyledons) of variants N90 and N120 were really high. Succinate dehydrogenase (SDH) is a key enzyme in the Krebs cycle that is part of cellular respiration [45]. Significant positive effect on SDH activity was noticed for soybean seeds treated with CAPP generated in oxygen atmosphere and ambient air. CAPP generated in a nitrogen atmosphere had a significant negative impact on SDH activity with increasing dose (Figure 6). In N90 and N120 variants, there was probably no switch from an anoxygenic to oxygenic metabolism, and the embryo suffocated. However, at lower doses (N30, N60), despite the negative effect on SDH activity, there was no statistically significant negative effect on germination (Figure 2). Germinating plants are clearly able to cope effectively with this level of stress. As our results show, the application of plasma generated in ambient air and oxygen atmosphere applied in short time interval can positively influence the metabolism (germination) of soybean by enhancement of hydrolytic enzymes. This is also supported by findings that argon plasma promotes soybean seed germination and the growth and development of young seedlings by regulating demethylation levels of energy metabolism-related genes [46].

To assess the DNA damage in seedlings pre-treated with CAPP, firstly the alkaline comet assay was used. This variant, however, was not suitable for our soybean samples, despite being used for pea seedlings (belonging to the same family as soybean) in our laboratory before [23]. The reason was very high DNA damage detected in the negative control (NC), which was close to the level of DNA damage in the positive control (PC). Although there was a difference between these samples the evaluation of such comets would be very challenging and probably distorted. In Figure 7, an example of the appearance of nucleoids in the negative (Figure 7A) and positive control (Figure 7B) after the alkaline comet assay is shown. Such a result of the alkaline comet assay might be associated with the age of seeds. Similar observations, although on bean leaves, have been reported by [47]. They observed an increase in DNA damage with increasing age of leaves using the alkaline comet assay. However, for these authors, possible explanations included reduced repair capacity in differentiated cells or sensitivity to alkali related to the degree of chromatin condensation [47]. The problem may indeed be with our soybean seeds, since the alkaline comet assay has successfully been used on soybean by several authors [48,49].

Subsequently, we decided to use the neutral comet assay to detect DNA breaks. The amount of DNA damage in the negative control was significantly lower than in alkaline version but still, it reached almost 20%. Such a high degree of DNA damage in NC may be associated with lower germination rate of our seeds. On the other hand, the positive control was only around 40%, as can be seen in Figure 8. A relatively low level of DNA damage in the neutral comet assay was also observed by Menke et al. [50] when using bleomycin (which belongs to the same family of antibiotics as zeocin) to induce DNA damage. They also observed rapid repair of DNA breaks caused by bleomycin, which might be the reason for such a low level of DNA damage detected for bleomycin and zeocin.

For seedlings pre-treated with CAPP generated in ambient air, an amount of DNA damage similar to NC, but with a slight increment with increasing treatment time, was observed, which proved to be statistically significant, as can be seen in Figure 8. Our observations are similar to those presented in [23]. They did not observe any significant changes in the amount of DNA damage in pea seedlings pre-treated with ambient air-CAPP (Diffuse Coplanar Surface Barrier Discharge (DCSBD); 400 W, 14 kHz) compared to the NC. The DNA damage of soybean seedlings pre-treated with CAPP generated in oxygen was similar to that for ambient air-CAPP; however, the highest level was in 60 s treatment time. There were significant differences for the treatment times of 60, 90 and 120 s, compared to NC (Figure 8). This is also in accordance with [6], where seedlings pre-treated with oxygen CAPP (DCSBD; 400 W, 14 kHz) had higher level of DNA damage compared to ambient air CAPP treatment. CAPP generated in nitrogen caused the highest DNA damage from all working gases studied in this work. DNA damage caused by nitrogen-CAPP was similar to PC even for the lowest treatment time (30 s) (Figure 8). There is no data for longer treatment times (90 and 120 s) because these samples did not germinate. Nitrogen-CAPP treatment also caused the highest DNA damage in pea seedlings in the study by [6], where an inhibition of germination for higher treatment times (180 and 300 s) was also observed. The DNA damage detected by the comet assay is a primary DNA damage; typically DNA breaks that can be repaired relatively quickly. As mentioned above, Menke et al. [50] observed a repair of bleomycin-induced DNA damage within an hour. If the repair of primary DNA damage is rapid, it has no negative effect on the vigour of seeds and young seedlings (Figure 2). Kyzek et al. [23] observed that CAPP-pretreatment of seeds may even mitigate the extent of DNA damage caused by zeocin. Thus, organisms are probably able to handle even slightly higher levels of DNA damage detected by Comet assay.

To detect more types of DNA damage by the neutral comet assay, a modification with repair enzyme Fpg to detect the presence of oxidized purines in nuclear DNA was added. There were small increases in DNA damage in almost every treatment time (except for oxygen-CAPP for 60 and 90 s) and working gas, meaning that CAPP-treatment may cause some oxidative DNA damage. The most prominent amount of oxidative damage was observed for 30 s treatment time for ambient air- and oxygen-CAPP (Figure 9).

## 3. Materials and Methods

### 3.1. Plant Material

Dried soybean (*Glycine max* L.) cv. Nížina seeds used in the experiments were obtained from the Central Control and Testing Institute in Agriculture in Bratislava, Slovakia in 2017. The seeds were stored in fridge at 8 °C in the dark. The experiments were performed in the years 2019 and 2020.

### 3.2. Plasma Source and Treatment of Soybean Seeds

For the plasma treatment of the soybean seeds, the Diffuse Coplanar Surface Barrier Discharge (DCSBD) [51] as a source of CAPP was used. DCSBD system is composed of many parallel electrodes situated from the bottom side of the ceramic (Al_2_O_3_), powered by high voltage sinusoidal signal (20 kV peak-to-peak) with frequency of 14 kHz and cooled by the cooling oil system. The plasma is created in macroscopically homogeneous thin layer on the ceramic surface. In our experiments, plasma was generated in ambient air (A), oxygen (O) and nitrogen (N) atmosphere at atmospheric pressure at the input power of 400 W. The seeds were put into the plasma for the required exposure time (0 s—control, 30 s—A30, O30, N30; 60 s—A60, O60, N60; 90 s—A90, O90, N90; 120 s—A120, O120, N120) using the experimental set-up in Figure 10. In case of nitrogen and oxygen, the plasma was situated in the cover into which the gas was flowed at the rate of 3 L/min. For the purpose of homogeneous plasma treatment, the seeds were moved by using the orbital shaker at the frequency of 270 rpm. The seeds were thus exposed to the plasma composed of many different reactive oxygen and nitrogen species, UV-radiation and other particles, which were studied in more detail by [6].

### 3.3. Imbibition, Germination and Growth Conditions

Dry soybean seeds (50 seeds for each variant) were weighed on analytical scales and soaked in sterile distilled water for 1 h at room temperature. Imbibed seeds were blotted dry, weighed again and wrapped in wet sterile filter paper. Rolls were cultivated in dark conditions in incubator at the temperature 24 ± 2 °C for 5 days. During cultivation the number of germinated seeds were counted and after three days the material for biochemical analyses was collected (Total soluble protein content, Assay on lytic enzymes assessment, Assay on antioxidant enzymes and visualization of ROS, Assay on dehydrogenase activities evaluation, Comet assay). After 5 days, the length and weight of shoots and roots of young seedlings were measured. We used these data to calculate Percentage of Germination (%), Germination Potential (%), Seed Vigour Index (%), Seedling Vigour Index (%), Seedling Length Index (%) according to [35].

### 3.4. Total Soluble Protein Content

Samples (~1.5 g) were ground in liquid nitrogen with mortar and pestle and suspended in 50 mM Na-Phosphate protein extraction buffer with 1 mM EDTA, pH 7.8. After 15 min centrifugation (12,000× *g*), the supernatant was used for determination of protein concentration according to Bradford [43]. Total soluble protein content was calculated as amount of total protein per gram of fresh matter from the calibration curve. As protein standard, Bovine Serum Albumin (BSA) was used.

### 3.5. Assay on Lytic Enzymes Assessment

The activity of β-1,3-Glucanase was assayed by measuring the rate of release of reducing sugar from laminarin (Sigma-Aldrich Co., Bratislava, Slovakia) as a substrate according to methodology by [52,53]. Absorbance was measured by a spectrophotometer Jenway 6705 UV/Vis (Bibby Scientific Ltd., Essex, UK) at 660 nm. Total enzyme activity was calculated from calibration curve. Glucose was used as a standard.

Changes in activity of protease in 3-day-old soybean seedling were determined by incubating 150 µL of an extracted protein sample with 150 µL of 2% (*w*/*v*) Bovine Serum Albumin (BSA) in 200 mM glycine-HCl (pH 3.0) at 37 °C for 1 h. The reaction was stopped by the addition of 450 µL of 5% (*w*/*v*) trichloroacetic acid. Samples were incubated on ice for 10 min, and centrifuged at 20,000× *g* for 10 min at 4 °C. Absorbance of the supernatant at 280 nm was measured by a spectrophotometer Jenway 6705 UV/Vis (Bibby Scientific Ltd., Essex, UK, [54]).

For determination of α-Amylase activities, a commercially available colorimetric assay kit purchased from Sigma-Aldrich Co. was used. One unit of α-amylase activity is the amount of amylase that cleaves ethylidene-pNP-G7 to generate 1.0 mM of p-nitrophenol per minute at 25 °C.

### 3.6. Assay on Antioxidant Enzymes, Guaiacol Peroxidase (POX, E.C.1.11.1.7) and Superoxide Dismutase (SOD, E.C.1.15.1.1) Activities Assessment, and Visualization of ROS (H_2_O_2_ and ˙O_2_^−^)

The activity of enzymes that detoxify ˙O_2_^−^ (SOD, E.C.1.15.1.1) and H_2_O_2_ (POX, E.C.1.11.1.7) was tested. The activity of superoxide dismutase was established according to [55] and the guaiacol peroxidase according to [56]. One unit of SOD activity is the amount of proteins required to inhibit 50% of initial reduction of Nitrotetrazolium Blue chloride (NBT) under the light. Guaiacol peroxidase activity is expressed in μM of tetraguaiacol min^−1^.mg^−1^ by molar extinction coefficient of tetraguaiacol 26.6. Chemicals were purchased from Sigma-Aldrich Co. The presence of H_2_O_2_ and ˙O_2_¯ was detected in 3-day-old soybean seedling according to [57]. H_2_O_2_ was visualized as a reddish brown stain formed by the reaction of 3,3′-Diaminobenzidine (DAB) with the endogenous H_2_O_2_. The content of ˙O_2_^−^ was detected as dark blue stain of formazan compound formed as the result of NBT reacting with the endogenous ˙O_2_¯.

### 3.7. Assay on Dehydrogenase Activity Evaluation

For determination of Alcohol (ADH), Lactate (LDH) and Succinate (SDH) dehydrogenases in 3-day-old seedlings, commercially available colorimetric assay kits purchased from Sigma-Aldrich Co. were used. Activities of enzymes were determined according to manufacturer’s instructions. One unit of ADH represents the amount of enzyme that will generate 1.0 mM of NADH per minute at pH 8.0 at 37 °C. One unit of LDH activity is defined as the amount of enzyme that catalyses the conversion of lactate into pyruvate to generate 1.0 µM of NADH per minute at 37 °C. One unit of SDH is the amount of enzyme that generates 1.0 µM of 2, 6-dichlorophenolindophenol (DCIP) per minute at pH 7.2 at 25 °C.

### 3.8. Comet Assay

The comet assay, with its many modifications, can be used to detect single- and double strand breaks, apyrimidinic and apurinic sites, base damage, etc. [58,59]. Alkaline comet assay was performed according to [23]. Briefly, nuclei were isolated from the roots of 3-day-old soybean seedlings using slicing method with a sharp razorblade in cold 0.4 M Tris-HCl set on ice. The nuclei fixed in agarose on the microscopic slide were left to unwind in a cold electrophoretic buffer (0.3 M NaOH (Centralchem, Banská Bystrica, Slovakia), 1 mM EDTA (Sigma-Aldrich Co.), pH > 13) for 8 min. Subsequently, electrophoresis was performed in the same buffer for 15 min at 1 V/cm and 4 °C. Slides were then neutralized with 0.4 M Tris-HCl pH 7.5 (Sigma-Aldrich Co.) buffer and stained with 15 μL of 0.05 mM ethidium bromide (Serva, Heidelberg, Germany) for each mini gel on the slide. The stained nuclei were observed using fluorescence microscope OLYMPUS BX 51 with a green excitation filter UMWIG3 under 200× magnification and at least 100 nuclei per slide were evaluated by the visual scoring method [58].

A neutral version of the comet assay was also performed. This version does not use a high alkali solution to unwind the DNA strands and, therefore, it detects only original strand breaks [58]. Due to the lack of alkaline conditions, it was not possible to detect alkali labile sites. The neutral comet assay was performed according to [50] with some modifications. The nuclei were isolated under the dim light from the roots of 3-day-old soybean seedlings using slicing method with a sharp razorblade in 150 µL of cold 1x PBS (160 mM NaCl (Centralchem), 8 mM Na_2_HPO_4_ (Centralchem), 4 mM NaH_2_PO_4_ (Centralchem), 50 mM EDTA (Sigma-Aldrich Co.), pH 7) set on ice. For each sample two roots were used, approximately one centimetre from apex, but excluding the root apex. The suspension of nuclei (100 µL) was mixed with 100 µL of 1% low melting point agarose (LMP (Roth, Karlsruhe, Germany)), pipetted onto slides precoated with 1% normal melting point agarose (Roth), and covered with coverslip. The nuclei fixed in LMP agarose on microscopic slides were lysed in lysis buffer (2.5 M NaCl (Centralchem), 100 mM EDTA (Sigma-Aldrich Co.), 10 mM Tris-HCl pH 7.5 (Sigma-Aldrich Co.), with 1% N-lauroylsarcosine sodium salt (Applichem, Darmstadt, Germany) and 1% Triton X-100 (Sigma-Aldrich Co.) added just before use) for 20 min. Subsequently, electrophoresis was performed in 1x TBE (90 mM Tris-HCl pH 8.4 (Sigma-Aldrich Co.), 90 mM Boric acid (Sigma-Aldrich Co.), 2 mM EDTA (Sigma-Aldrich Co.)) at 0.7 V/cm at room temperature for 10 min. The slides were then dehydrated for 5 min in 75% ethanol and 5 min in 96% ethanol, left to dry and stored for subsequent evaluation. Nuclei were stained, observed, and evaluated in the same way as in the alkaline version of comet assay.

The comet assay has several modifications to detect more types of DNA damage. One of them uses a repair enzyme formamidopyrimidine DNA glycosylase (Fpg (New England Biolabs, Ipswich, MA, USA)). Fpg removes oxidized purines and creates a break in a DNA strand which can be detected by the comet assay [60,61]. The Fpg-modified comet assay was based on a neutral comet assay described before with the Fpg additional steps according to [62]. Briefly, after the lysis step, the slides were washed three times for 5 min with an enzyme reaction buffer (40 mM HEPES (Applichem), 0.5 mM EDTA (Sigma-Aldrich Co.), 0.1 M KCl (Sigma-Aldrich Co.), 0.2 mg/mL BSA (Sigma-Aldrich Co.), pH 8). Then, 0.2 U (40 μL) of enzyme was added to each mini gel on slide and covered with a coverslip. The same amount of enzyme reaction buffer (instead of Fpg) was added to a parallel series of samples and covered with a coverslip. Both series were placed into a thermostat at 37 °C for 30 min. After the incubation, coverslips were removed, and the protocol continued with the step of electrophoresis as in the standard neutral comet assay.

A radiomimetic antibiotic zeocin (Invivogen, Toulouse, France) (5 mg/mL, 1 h) was used to treat seedlings in a positive control (PC) to induce DNA breaks [63] in neutral version of Comet assay. In alkaline version, H_2_O_2_ (450 µM) (Sigma-Aldrich Co.) was used to induce DNA damage. As a negative control (NC), seedlings not pre-treated with CAPP or any other agent were used.

### 3.9. Statistical Analysis

The data were analysed using Microsoft Excel (Microsoft Office 2013) and Statgraphic Centurion 19 (Statgraphics Technologies, Inc. The Plains, Virginia). Treatment effects were investigated by means of ANOVA single-step multiple comparisons of means by means of LSD tests or Tukey’s HSD test and comparisons between the mean values were considered significant at *p* < 0.05. All experimental data in this work are from at least three independent experiments.

## 4. Conclusions

The effect of CAPP on soybean seeds germination, growth and development and the antioxidant response of young seedlings depends on its dose, but especially on the used working gas. In general, we can state that all treatments (except N90 and N120) had a positive effect on germination (Percentage of Germination (%), Germination Potential (%)). In the case of the O60 variant, the seed germination increased up to 80% compared to the untreated control (60%). This is probably due to the positive effect of CAPP on activity of enzymes, which are essential in initial germination phases (cleavage and mobilization of stock substances from cotyledons—proteases, amylase, glucanase; switching from anoxygenic metabolism to oxygenic—activities of lactate and alcohol dehydrogenases vs. succinate dehydrogenase). As the reduced germination of untreated soybean seeds is attributed to the age of the seeds, the increase in the germination due to the CAPP treatment is an important finding for agricultural practice. High doses of CAPP generated in a nitrogen atmosphere (N90, N120) caused significant oxidative stress (high accumulation of ˙O_2_^−^ in cotyledons), which led to complete inhibition of germination via pathological ROS production, lipid peroxidation and failure of enzymatic and non-enzymatic detoxification defence mechanisms, which may be associated with the production of lactate and ethanol and with the high activity of the LDH and ADH enzymes. We attribute the accumulation of ˙O_2_^−^ and H_2_O_2_ in the root tips and the elongation zone of the roots of soybean seedlings in other treatments with intensive cell division and elongation. The low level of stress is also confirmed by the positive effect of CAPP on the seedling length (cm) and seedling length index (%), compared to the untreated control. The amount of DNA damage slightly increased with increasing treatment times in every working gas; however, there were also differences between working gases. CAPP generated in ambient air caused the least DNA damage, slightly above the level of NC. On the other hand, CAPP generated in nitrogen caused the DNA damage similar to the PC, which was caused by antibiotic zeocin, even at the shortest treatment times (30 and 60 s). The CAPP treatment of soybean seeds probably does not cause the excessive oxidative damage of DNA since we detected only small amounts of this type of DNA lesions.

## Figures and Tables

**Figure 1 plants-10-00177-f001:**
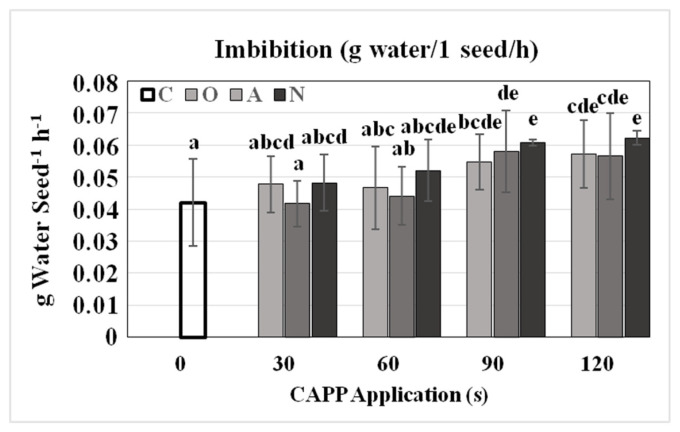
Imbibition rate at room temperature by soybean seeds treated and untreated (control) by CAPP. Variants: C—control/untreated soybean seeds; O30, O60, O90, O120—soybean seeds treated with plasma generated in oxygen atmosphere for 30, 60, 90 or 120 s; A30, A60, A90, A120—soybean seeds treated with plasma generated in ambient air for 30, 60, 90 or 120 s; N30, N60, N90, N120—soybean seeds treated with plasma generated in nitrogen atmosphere for 30, 60, 90 or 120 s. Different letters indicate significant difference at *p*-value < 0.05, bars are means of four experimental runs (1 run represents 50 seeds per variant, n = 200) ± SD according to Tukey’s HSD test.

**Figure 2 plants-10-00177-f002:**
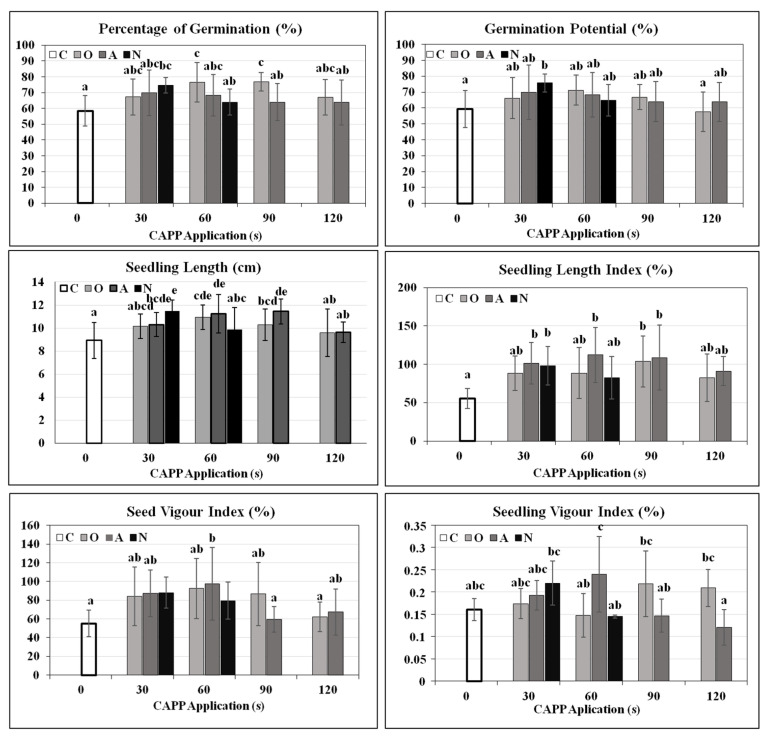
Percentage of Germination (%), Germination Potential (%), Seedling Length (cm), Seedling Length Index (%), Seed Vigour Index (%) and Seedling Vigour Index (%) of soybean seeds of 5-day-old soybean seedlings after CAPP treatment. Variants: C—control/untreated soybean seeds; O30, O60, O90, O120—soybean seeds treated with plasma generated in oxygen atmosphere for 30, 60, 90 or 120 s; A30, A60, A90, A120—soybean seeds treated with plasma generated in ambient air for 30, 60, 90 or 120 s; N30, N60, N90, N120—soybean seeds treated with plasma generated in nitrogen atmosphere for 30, 60, 90 or 120 s. Different letters indicate significant difference at *p*-value < 0.05, bars are means of four experimental runs (one run represents 50 seeds per variant, n = 200) ± SD according to Tukey’s HSD test. Indexes were calculated according to [35], Percentage of germination (%) = (number of germinated seeds/total number of seeds) × 100; Germination potential (%) = (number of germinated seeds in 5 days/total number of seeds) × 100; Seedling length index (%) = Germination index × total length of 5-day-old seedlings; Seed Vigour index (%) = ((length of roots of 5-day-old seedlings + length of shoots of 5-day-old seedlings in mm) × Percentage of germination)/100; Seedling Vigour index (%) = ((weight of roots of 5-day-old seedlings + weight of shoots of 5-day-old seedlings in g) × Percentage of germination)/100. Length of seedling (cm) = 5-day-old seedlings were measured.

**Figure 3 plants-10-00177-f003:**
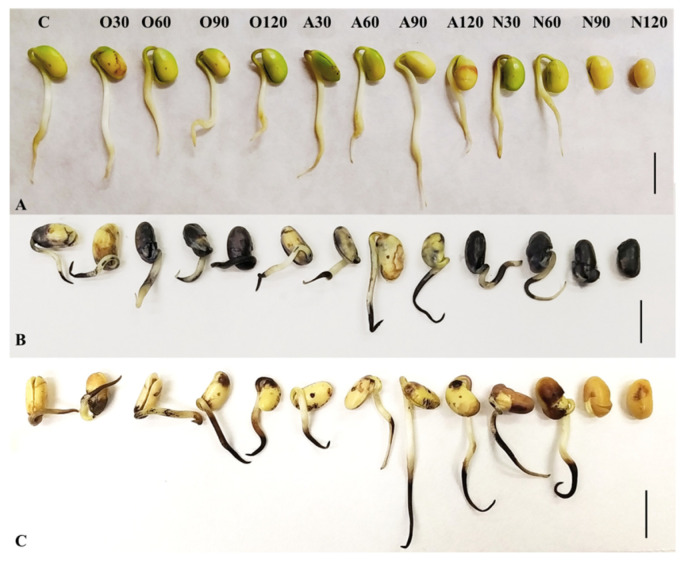
Three-days old soybean seedling after CAPP treatment (**A**), Superoxide (**B**) and hydrogen peroxide (**C**) visualisation in 3-day-old soybean seedlings by using DAB and NBT after CAPP treatment. The presence of ˙O_2_^−^ and H_2_O_2_ is confirmed by blue or brownish spots, respectively. Variants: C—control/untreated soybean seeds; O30, O60, O90, O120—soybean seeds treated with plasma generated in oxygen atmosphere for 30, 60, 90 or 120 s; A30, A60, A90, A120—soybean seeds treated with plasma generated in ambient air for 30, 60, 90 or 120 s; N30, N60, N90, N120—soybean seeds treated with plasma generated in nitrogen atmosphere for 30, 60, 90 or 120 s. Bar = 1 cm. Experiment was repeated four times (one run = five seedlings from each variant were incubated in NBT solution and 5 seedlings from each variant were incubated in DAB solution, n = 20). The image represents a representative selection of seedlings after staining.

**Figure 4 plants-10-00177-f004:**
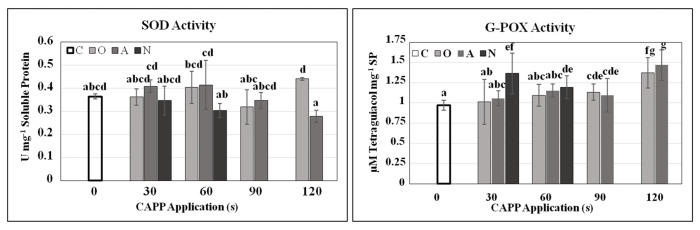
Activity of superoxide dismutase (SOD) and guaiacol peroxidases (G-POX) in 3-day-old soybean seedlings after CAPP treatment. Variants: C—control/untreated soybean seeds; O30, O60, O90, O120—soybean seeds treated with plasma generated in oxygen atmosphere for 30, 60, 90 or 120 s; A30, A60, A90, A120—soybean seeds treated with plasma generated in ambient air for 30, 60, 90 or 120 s; N30, N60, N90, N120—soybean seeds treated with plasma generated in nitrogen atmosphere for 30, 60, 90 or 120 s. Different letters indicate significant difference at *p*-value < 0.05, bars are means of four experimental runs (one run represents 50 seeds per variant; three 1.5 g mixed samples were analysed per experimental run and each variant) ± SD according to Tukey’s HSD test.

**Figure 5 plants-10-00177-f005:**
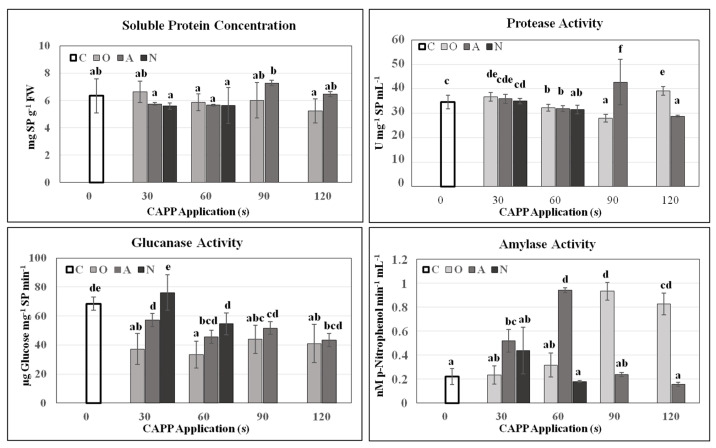
Total soluble protein content and activities of lytic enzymes (protease, glucanase, amylase) in 3-day-old soybean seedlings after CAPP treatment. Variants: C—control/untreated soybean seeds; O30, O60, O90, O120—soybean seeds treated with plasma generated in oxygen atmosphere for 30, 60, 90 or 120 s; A30, A60, A90, A120—soybean seeds treated with plasma generated in ambient air for 30, 60, 90 or 120 s; N30, N60, N90, N120—soybean seeds treated with plasma generated in nitrogen atmosphere for 30, 60, 90 or 120 s. Different letters indicate significant difference at *p*-value < 0.05, bars are means of four experimental runs (one run represents 50 seeds per variant; three 1.5 g mixed samples were analysed per 1 experimental run and each variant for soluble protein concentration, protease activity and glucanase activity; five 0.1 g mixed samples were analysed per 1 experimental run and each variant for amylase activity) ± SD according to Tukey’s HSD test.

**Figure 6 plants-10-00177-f006:**
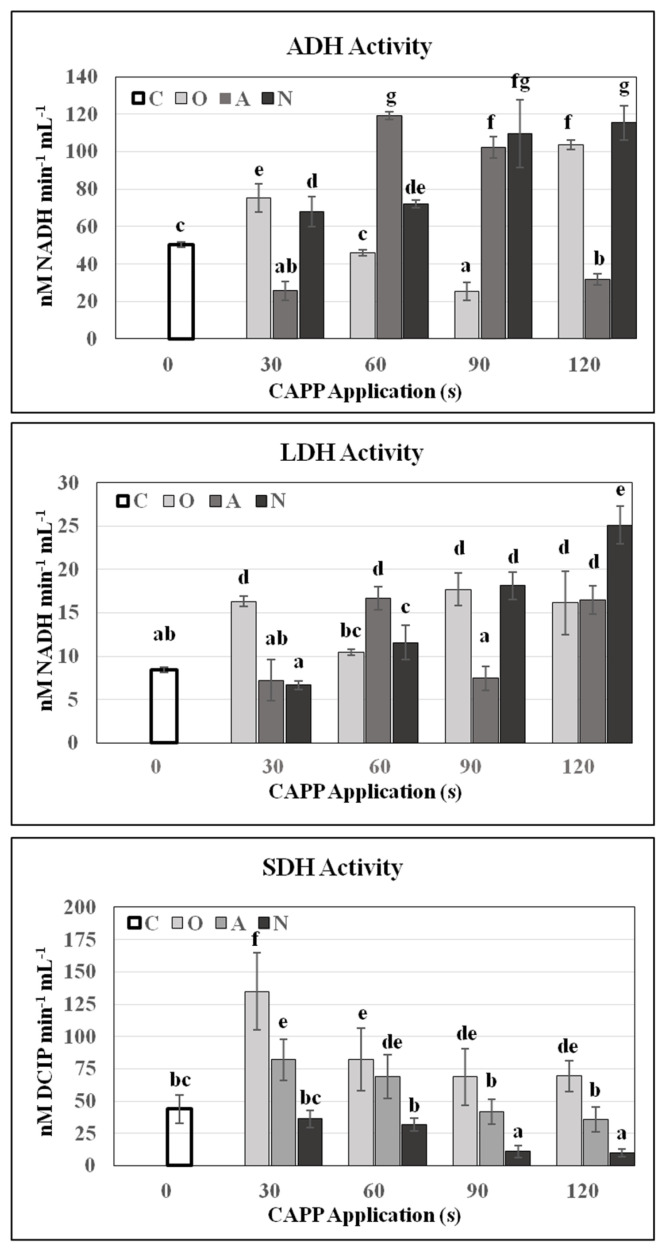
Activity of alcohol dehydrogenase (ADH), activity of lactate dehydrogenase (LDH) and activity of succinate dehydrogenase (SDH) in 3-day-old soybean seedling after CAPP treatment. In variants N90 and N120, activities of ADH, LDH and SDH were determined in imbibed seeds (embryo + cotyledons), because these seeds did not germinate. Variants: C—control/untreated soybean seeds; O30, O60, O90, O120—soybean seeds treated with plasma generated in oxygen atmosphere for 30, 60, 90 or 120 s; A30, A60, A90, A120—soybean seeds treated with plasma generated in ambient air for 30, 60, 90 or 120 s; N30, N60, N90, N120—soybean seeds treated with plasma generated in nitrogen atmosphere for 30, 60, 90 or 120 s. Different letters indicate significant difference at *p*-value < 0.05, bars are means of four experimental runs (one run represents 50 seeds per variant; five 0.1 g mixed samples were analysed per experimental run, and each variant for LDH and SDH activity; five 0.05 g mixed samples were analysed per experimental run, and each variant for ADH activity) ± SD according to Tukey’s HSD test.

**Figure 7 plants-10-00177-f007:**
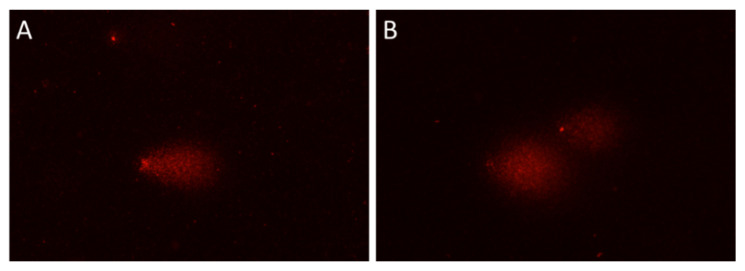
DNA damage in nucleoids of 3-day-old soybean seedlings detected by the alkaline comet assay. (**A**) Negative control (NC; no treatment); (**B**) positive control (PC; 450 μM H_2_O_2_, 45 min). DNA damage in NC is very high, almost all DNA is in the tail of the comet. The alkaline comet assay is not suitable for detection of DNA damage in these seedlings.

**Figure 8 plants-10-00177-f008:**
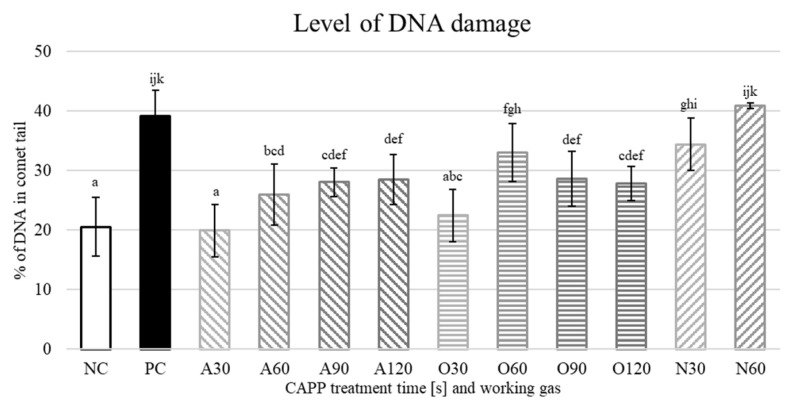
Level of DNA damage in 3-day-old soybean seedlings detected by the neutral comet assay. NC—negative control (no treatment); PC—positive control (zeocin 5 mg/mL, 1 h); A30, A60, A90, A120—seedlings pre-treated with CAPP generated in ambient air for 30—120 s; O30, O60, O90, O120—seedlings pre-treated with CAPP generated in oxygen for 30—120 s; N30, N60—seedlings pre-treated with CAPP generated in nitrogen for 30 and 60 s. At least 100 nuclei were analysed per slide. The data were analysed using statistical method LSD ANOVA and comparisons between the mean values were considered significant at *p* ≤ 0.05. Different letters indicate significant difference at *p*-value < 0.05, bars are means of five experimental runs.

**Figure 9 plants-10-00177-f009:**
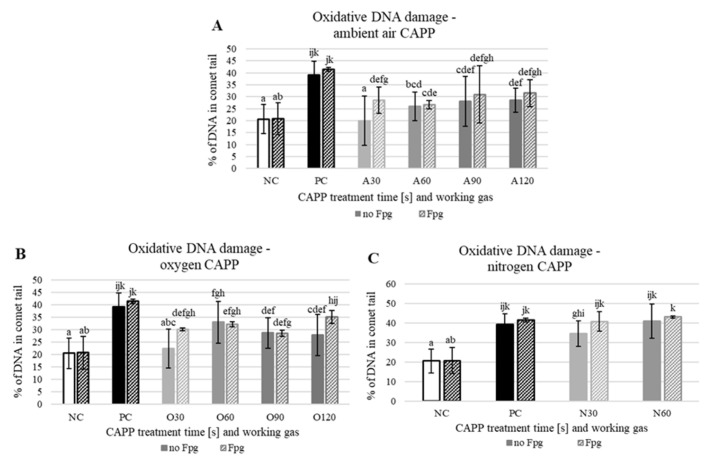
Level of DNA damage in 3-day-old soybean seedlings detected by the neutral comet assay modified with repair enzyme Fpg. NC—negative control (no treatment); PC—positive control (zeocin 5 mg/mL, 1 h); (**A**) A30, A60, A90, A120—Scheme 30–120 s; (**B**) O30, O60, O90, O120—seedlings pre-treated with CAPP generated in oxygen for 30–120 s; (**C**) N30, N60—seedlings pre-treated with CAPP generated in nitrogen for 30 and 60 s. At least 100 nuclei were analysed per slide. The data were analysed using statistical method LSD ANOVA and comparisons between the mean values were considered significant at *p* ≤ 0.05. Different letters indicate significant difference at *p*-value < 0.05, bars are means of four experimental runs.

**Figure 10 plants-10-00177-f010:**
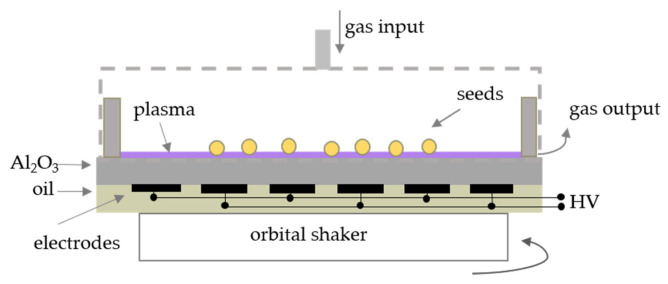
Scheme of the experimental set-up of the plasma treatment using DCSBD discharge.

## Data Availability

Data available in a publicly accessible repository.

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
