# Peer review of "Evaluation of the Impact of Cold Atmospheric Pressure Plasma on Soybean Seed Germination"

_plants, 2021, doi:10.3390/plants10010177_

Round 1
Reviewer 1 Report
2021-01-09
The manuscript “Evaluation of the impact of Cold Atmospheric Pressure Plasma on soybean seed germination” has been improved compared to the first version, however it still needs further improvements (see below). The novelty of this study is that authors aimed to find out the optimal protocol for treatment using comparison of the effects of CAPP generated in different gaseous atmosphere on numerous structural, physiological and biochemical parameters of seeds and seedlings.
Remarks and suggestions for improvement:
The abstract was rewritten but recommendation is to change it using the passive voice. Such “lonely sentences” as “Activity of protease slightly increased” should be avoided.
Introduction
- It is necessary to provide the reference for statista.com directing to the information relevant to the statement.
- Numerous reviews were published recently (2019-2020) providing an updated information of plasma applications in agriculture – their references should be given instead of [1].
- The description of negative and positive impacts of seed treatments with plasma should present more clear picture on factors that might responsible for this difference both from treatment and plant side.
- Some formulations in the end of introduction should be improved: e.g. “Our work provides unique knowledge about…” should be replaced by e.g. “This study reports novel data on…” The statement about effects on DNA deserves separate sentence with specified contribution of nitrogen atmosphere.
Results and discussion
- The sentence (on lines 98-100) “Study of [4] confirmed that the generation of plasma in a nitrogen and ambient air atmosphere produces ultraviolet (UV) radiation, which is harmful in high doses (according to WHO).” sounds strange because of two reasons: 1) generation of UV radiation during plasma discharge in different gaseous atmospheres was demonstrated much earlier by other authors [e.g., Deng et al., 2013, http://dx.doi.org/10.1063/1.4774328; Sureshkumar et al., 2010 doi:10.1016/j.ijpharm.2010.05.045]; (2) the indication to WHO is hardly relevant discussing plant DNA damage. Instead, the references on genotoxic UV effects on plants should be provided.
- Although English language was corrected, there are still numerous cases of inappropriate use of plural forms. Replace such phrases as seedlings length or seedlings vitality to seedling length, etc. (p. 3-4) or (total amount of) proteins to protein, etc. (p. 6)
Author Response
The abstract was rewritten but recommendation is to change it using the passive voice. Such “lonely sentences” as “Activity of protease slightly increased” should be avoided.
- the abstract has been rewritten
Introduction
It is necessary to provide the reference for statista.com directing to the information relevant to the statement.
- the reference for statista.com was added (line 34)
Numerous reviews were published recently (2019-2020) providing an updated information of plasma applications in agriculture – their references should be given instead of [1].
- we have added a two review articles from 2020:
[2] Ranieri, P.; Sponsel, N.; Kizer, J.; Rojas-Pierce, M.; Hernández, R.; Gatiboni, L.; Grunden, A.; Stapelmann, K. Plasma agriculture: Review from the perspective of the plant and its ecosystem,” Plasma Processe and Polymers. 2020, e2000162.
[3] Attri P.; Ishikawa, K.; Okumura, T.; Koga, K.; Shiratani,M. Plasma agriculture from laboratory to farm: A review. Processes. 2020, 8(8), 1002.
The description of negative and positive impacts of seed treatments with plasma should present more clear picture on factors that might responsible for this difference both from treatment and plant side.
- In an effort to provide a clearer picture of the factors that influence the responses of seed/plant germination to plasma treatment (whether positively or negatively), we have included in the Introduction citations documented that:
- the degree of damage/hardness of the seeds is affected by moisture in plant seed material and by seed coat thickness,
- type of plasma source, exposure time, and both factors together significantly affect seed/grain germination and growth of young seedlings and the plasma effect is also species specific.
[15] Frączek, J.; Hebda, T.; Ślipek, Z.; Kurpaska, S. Effect of seed coat thickness on seed hardness. Can. Biosyst. Eng. 2005, 47, 4.1–4.5.
[24] Šerá, B.; Gajdová, I.; Černák, M.; Gavril, B.; Hnatiuc, E.; Kováčik, D.; Kříha, V.; Sláma, J.; Šerý, M.; Špatenka, P. How various plasma sources may affect seed germination and growth. IEEE Meeting Conference Paper, May 2012, DOI: 10.1109/OPTIM.2012.6231880.
Some formulations in the end of introduction should be improved: e.g. “Our work provides unique knowledge about…” should be replaced by e.g. “This study reports novel data on…” The statement about effects on DNA deserves separate sentence with specified contribution of nitrogen atmosphere.
- “Our work provides unique knowledge about…” has been replaced by “This study reports novel data on…” (line 80). The statement about effects on DNA has been expanded (lines 82-85).
Results and discussion
The sentence (on lines 98-100) “Study of [4] confirmed that the generation of plasma in a nitrogen and ambient air atmosphere produces ultraviolet (UV) radiation, which is harmful in high doses (according to WHO).” sounds strange because of two reasons: 1) generation of UV radiation during plasma discharge in different gaseous atmospheres was demonstrated much earlier by other authors [e.g., Deng et al., 2013, http://dx.doi.org/10.1063/1.4774328; Sureshkumar et al., 2010 doi:10.1016/j.ijpharm.2010.05.045]; (2) the indication to WHO is hardly relevant discussing plant DNA damage. Instead, the references on genotoxic UV effects on plants should be provided.
- The above sentence concerned DCSBD plasma, which we used also in our work, so we specified it and added the referrence about the UV effect on plants as follows:
- Study of [6] confirmed that the generation of DCSBD plasma, used also in our work, in a nitrogen and ambient air atmosphere produces ultraviolet (UV) radiation, which is harmful in high doses as confirmed by the work [29]. (lines 113-115).
[29] Gill, S.S.; Anjum, N.A.; Gill, R.; Jha, M.; Tuteja, N. DNA damage and repair in plants under ultraviolet and ionizing radiations. Sci. World J. 2015, https://doi.org/10.1155/2015/250158.
Although English language was corrected, there are still numerous cases of inappropriate use of plural forms. Replace such phrases as seedlings length or seedlings vitality to seedling length, etc. (p. 3-4) or (total amount of) proteins to protein, etc. (p. 6)
- The English was checked again by a native speaker. Plural forms have been corrected (in Figures, legends and in text).
Reviewer 2 Report
Thank you for the revision. I hope that this technique will have large prospects in agriculture
Author Response
The reviewer had no questions or comments.
Reviewer 3 Report
No comments
Author Response
Reviewer had no no questions.
- The English was checked again by a native speaker.
This manuscript is a resubmission of an earlier submission. The following is a list of the peer review reports and author responses from that submission.
Round 1
Reviewer 1 Report
The study “Cold Atmospheric Pressure Plasma has a positive effect on soybean seed germination” is one of numerous studies published on this topic. The first reported case of plasma application to seeds (in a US patent by Krapivina et al. in 1994) was based on the observed strong positive effects of plasma on soybean germination and seedling growth. The novelty of this study is that authors aimed to find out the optimal protocol for treatment using comparison of the effects of CAPP generated in different gaseous atmosphere on numerous structural, physiological and biochemical parameters of seeds and seedlings. Although this study provides a potentially important and interesting set of novel data, the quality of manuscript preparation, description of used methods, presentation of results, discussion and formulation of sentences must be substantially improved. In general, the presentation of study can be characterized as careless. In addition, the correction of English by the language expert is absolutely required.
Remarks and suggestions for improvement:
The introduction is rather messy and has to be rewritten, particularly the final part.
Contradictions are clear between some sentences in introduction. E.g. authors claim that “The positive or negative impact on germination, growth and development is species specific and strongly depends on type of plasma and time of application [14].” The next statement is: “Based on the results of our previous study on pre-sowing treatment of pea seeds using CAPP 82 [14], in this study, we have reduced the CAPP exposure times on soybean seeds to 30 – 120 s.“ Thus, authors base a protocol for soybean seed treatment on the results obtained on pea seeds, despite obvious dependence of CAPP effects on plant species?
Authors believe that effects pf CAPP on germination can be explained by the induced changes in water uptake. How to explain then absence of correlation between imbibition rate and germination indices for CAPP application N30, N90, N120?
The is statement (on p. 3) “Study of [7] confirmed that the generation of plasma in a nitrogen atmosphere produces ultraviolet radiation, which is harmful in high doses (according to WHO)” needs explanation – is UV not generated in A or O atmosphere? Or it is less dangerous?
Figure 2: Seed germination and seedling growth are characterized using indices described in reference [46], which was published in 1973. The explanation on the used parameters should be provided for readers. It is a bit strange way to present the data, e.g., when length index (%) is used instead seedling length. Why green colour is used in for C symbol, but is not used for the column? It is indicated in the legend that bars are means of 4 experimental runs ± SD – what is experimental run in this case?
The number of repetitions of the germination test and the number of seeds used for one repetition should be indicated, as well as the number of seedlings used for length measurement (or it is n=4?). The same remark is for the most of illustrations.
Figure 3: The experimental groups are marked by using 0 (zero) instead O, and V - instead A? The legend says: Different letters indicate significant difference at P < 0.05, bars are means of 4 experimental runs ± SD according to LSD ANOVA test. How can you explain that? Looking to the length of not stained and stained seedlings belonging to the same group, it is obvious, that accidentally selected individual seedlings cannot represent the group – indeed some statistical analysis is needed there. The differences in the length of selected seedlings hardly corresponds to the results on Figure 2.
Figure 7 and Figure 9: Based on the results of the performed Comet assay, in this and in several earlier papers authors claim that exposure of plant seeds to CAPP induces strong DNA damage. However, strong genotoxic effects do not support statements about eco-friendly nature of such treatments and rises doubts the idea of using plasma for agriculture, in general. Therefore, genotoxic effects must be proven by a very solid evidence. However, the reliability of this assay seems quite suspicious due to high level of DNA damage detected in the negative control. That indicates that something is wrong with the applied protocol. However, the description of the Comet assay is not provided in a sufficient detail in this manuscript as well as in the cited reference [24] (which is published not in a biological journal). E.g. the method and intensity of razor blending must be described because root chopping procedure can induce DNA damage. The number of analyzed nuclei per slide should be indicated also, since it is critical to avoid DNA damage using Comet assay protocol, as noted by Pourrut et al., Mutagenesis 2015, 30, 37-43. Explanation is needed what is positive and negative control. DNA damage in negative control was ~5 %, in positive control - >90% in roots of V. faba and 3 other plant species (Pourrut et al., 2015). In this study the negative control was ~20% , and positive control - <40% despite 9 times larger concentration (450 mM) of H2O2 used compared to Pourrut et al., 2015. Although statistical significance is indicated, the number of experimental repetitions and number of the analysed nuclei is not indicated. Authors are happy about the presence of statistically significant difference (20%) between negative and positive control, however such small difference indicates poor quality of this experiment.
Referencing: Selection of references looks rather strange (the suggestion is to replace some of references (4, 5,6) by the more appropriate references to support statements. e.g., Lu et. Al. (2016). Physics Reports, 630, 1–84.). It is claimed (p.2): “Application of non-thermal plasma generated in ambient air expressively influenced, due to the surface oxidation, the wettability and water uptake by wheat, oat, lens and bean seeds [12].”: however, oat seeds were not used for the cited study. The same wrong statement is repeated on p. 3. The surnames are confused with first names of authors for Reference [28].
Numerous similar remarks, indicating insufficient quality of the submitted manuscript, can be given for the rest part of the manuscript. However, the arguments mentioned above are sufficient to substantiate the suggestion to reject the submitted paper.
Reviewer 2 Report
Minor comments:
- At the beginning of the Introduction section the authors enumerate the main benefits of soybean seeds (line 34). Among the former they list phytic acid and dietary elements. But phytic acid is considered to be an anti-nutrient. Furthermore, accumulation levels of selenium in soybean is high only in case of high selenium concentration is soil. May be it will be better only to indicate that like other legumes soybean may accumulation high levels of selenium. Besides nothing is said about high ability of soybean to concentrate molybdenum. More important are high levels of protein and lipids in soybean seeds.
- It is necessary to check all the abbreviations in the text. In several cases some abbreviations are not deciphered (for instance, NBT, DAB, DSA). That may be connected with the fact that Material and Methods section is situated at the end of the manuscript. Figure 5 should contain full title of soluble protein but not SP. Fig.7- please give full title of NC, PC. It is not clear was is negative and positive control
Reviewer 3 Report
Comments to the Author
I have carefully reviewed your manuscript: "Cold Atmospheric Pressure Plasma has a positive effect on soybean seed germination". This work describes fundamental research of a CAPP technique generated by different gases to find out the most suitable exposure time of non-thermal plasma pre-sowing treatment, which will be most suitable in relation to the enhancement of seed germination and overall growth parameters. Although I found merit in your study, I have a number of serious concerns that preclude acceptance in its present form.
Comments to the Author
x Change the title to be " Evaluation the impact of Cold Atmospheric Pressure Plasma on soybean seed germination" (you don't know if this technique has a positive effect, you try to prove that)
x Re-write your abstract to be a real scientific abstract. you wrote your abstract as an introduction. Your abstract should be a summery for your idea, methods, results, and conclusion.
x The introduction should address the novelty of this study. In the last paragraph in your introduction you mentioned some results and you discussed. These information have to move to results and discussion part.
x at the material and methods a scheme of the equipment should be added.
x at the results and discussion, comments in a critical way with the relevance and significance of the results obtained.
x How is the limitation for adoption of cold plasma at industrial level? Please comment.